# The Application of *Moringa oleifera* Leaf Meal and Its Fermentation Products in the Diet of *Megalobrama amblycephala* Juveniles

Wenqiang Jiang [1], Linjie Qian [1], Yongfeng Zhao [1,2], Yan Lin [2], Yang Yang [1], Huaishun Shen [1,2,*,†], Xianping Ge [1,2] and Linghong Miao [1,2,*,†]

1   Wuxi Fisheries College, Nanjing Agricultural University, Wuxi 214081, China; 2020213006@stu.njau.edu.cn (W.J.); 2022213003@stu.njau.edu.cn (L.Q.); zhaoyf@ffrc.cn (Y.Z.); yangyang@geneplus.org.cn (Y.Y.); gexp@ffrc.cn (X.G.)

2   Key Laboratory of Freshwater Fisheries and Germplasm Resources Utilization, Ministry of Agriculture and Rural Affairs, Freshwater Fisheries Research Center, Chinese Academy of Fishery Sciences, Wuxi 214081, China; liny@ffrc.cn

*   Correspondence: shenhs@ffrc.cn (H.S.); miaolh@ffrc.cn (L.M.); Tel./Fax: +86-510-8555-8830 (H.S.); +86-510-8555-8830 (L.M.)

†   These authors contributed equally to this work.

**Abstract:** This study assessed the potential applicability of *Moringa oleifera* leaf meal (MO) and fermented *Moringa oleifera* leaf meal (FMO) as feed supplements for aquatic animals. Five experimental diets, including the basal diet (control), 2.2% and 4.4% MO-supplemented diets (MO2 and MO4), and 2.2% and 4.4% FMO-supplemented diets (FMO2 and FMO4), were prepared for feeding *Megalobrama amblycephala* juveniles. After the eight-week feeding trial, the growth performance, muscle nutritional composition, plasma and hepatic biochemistry indicators were measured. The results demonstrated that MO and FMO had no detrimental effects on the growth performance of *M. amblycephala* juveniles. The muscle crude protein, crude lipid, and total free amino acids contents were significantly enhanced in the FMO4 group ($p < 0.05$). The liver acetyl-CoA carboxylase α mRNA level was significantly increased and the lipoprotein lipase mRNA level was markedly reduced in the FMO4 group ($p < 0.05$). Therefore, the FMO4 group exhibited a significant increase in plasma low-density lipoprotein cholesterol and triglyceride levels ($p < 0.05$). Compared to the control group, total superoxide dismutase and catalase activities were significantly increased in the FMO4 group ($p < 0.05$). The FMO2 and FMO4 groups exhibited an anti-inflammatory response by inhibiting the expression levels of toll-like receptor 4, nuclear factor-kappa B, and tumor protein P53 mRNA ($p < 0.05$). In conclusion, the 4.4% FMO treatment increased muscle crude protein content, enhanced lipogenesis, and improved the hepatic antioxidant abilities of *M. amblycephala* juveniles, while 2.2% FMO and 4.4% FMO improved the liver anti-inflammatory capacities.

**Keywords:** *Moringa oleifera* leaf meal; fermentation; lipid metabolism; antioxidant; anti-inflammatory





## 1. Introduction

In recent years, resource shortages have led to elevated prices of fishmeal, soybean meal, and cottonseed meal in aquatic feed, as well as other protein sources [1]. To alleviate the phenomenon of competition between humans and animals for food, some non-food protein sources obtained widely concerned, including mycoprotein (*Clostridium autoethanogenum* protein, yeast hydrolysate) [2,3], insect protein (*Tenebrio molitor* meal, *Hermetia illucens* meal) [4,5], and plant leaf protein (mulberry leaf meal, *Hygrophila spinosa* leaf meal) [6,7]. In addition, some processing technologies, such as enzymatic digestion [8] and fermentation [6,9], were also used for improving the quality and utilization efficiency of raw feed ingredients by aquatic animals.

*Moringa oleifera* is abundant in tropical and subtropical regions. This versatile plant has diverse applications in agriculture, medicine, animal husbandry, and aquaculture due to its rich nutritional value and antioxidant capacity [10]. *M. oleifera* is used in some developing countries to supply nourishment and guarantee food security. *M. oleifera* leaf, the main waste material of *M. oleifera* agriculture, has been reported to contain greater proportions of vitamins C and A, calcium, potassium, iron, and protein compared to other foods such as milk, fruits, and vegetables [11]. *M. oleifera* leaf contains up to 27.6–35.4% of protein, with as many as 19 kinds of amino acids [12]. In addition to its rich nutritional composition, *M. oleifera* leaf also contains a wide array of active substances (e.g., kaempferol, isoquercitrin) with hypolipidemic, anti-inflammatory, antioxidant, hepatoprotective, and antibacterial properties [13–15].

Several studies have demonstrated that using *M. oleifera* leaf meal (MO) as a protein source in formulated diets had a positive effect on fish growth and immunity. For example, dietary supplementation with 1.5% MO promoted the development of Nile tilapia (*Oreochromis niloticus*) and alleviated the damage caused by starvation stress [16]. Furthermore, supplementation with 5% MO improved the systemic immunity (humoral and cellular immunity) of sea bass (*Lateolabrax japonicus*) [17]. In another study, dietary supplementation with 200 mg/kg of an *M. oleifera* aqueous extract was found to enhance growth performance and immune organ function in *O. niloticus* [18]. Other studies reported that the application of MO and *M. oleifera* leaf extracts enhanced the tissue antioxidant activities of *O. niloticus* and giant freshwater prawn (*Macrobrachium rosenbergii*) [13,19,20]. Nevertheless, MO can negatively affect the growth performance of aquatic animals (e.g., Bocourti's catfish (*Pangasius bocourti*), African catfish (*Clarias gariepinus*), and *O. niloticus*) when supplemented at levels >10% due to its content of anti-nutritional factors such as saponins and tannins, among others [21–24]. An early study reported that the nutritional quality and nutrient bioavailability of MO could be greatly enhanced through microbial fermentation processes [25]. Zhang et al. [9] reported that fermentation of MO dramatically increased the flavonoid and polysaccharide contents of the fermentation products by 11.18% and 17.39%, respectively. In turn, fermented *M. oleifera* leaf meal (FMO) improved the non-specific immunity and antioxidant capacity of Gibel carp (*Carassius auratus gibelio*) [9]. Furthermore, fermentation products promoted the growth performance of aquatic animals by affecting the intestinal environment [26,27]. Replacing 50% and 10% of fishmeal with fermented soy pulp and fermented rice protein, respectively, as protein sources in formulated diets improved the activities of digestive enzymes in *C. gariepinus* and hybrid grouper (*Epinephelus fuscoguttatus*♀× *E. lanceolatus*♂) [26,27]. However, very few studies have characterized the effects of MO on lipid metabolism in aquatic animals. MO has a hypolipidemic effect, inhibiting cholesterol synthesis and reducing hepatic lipid vacuolation in mono-sex tilapia (*Oreochromis niloticus*) [28]. *M. oleifera* leaf products effectively prevented hypercholesterolemia and lipid deposition in mice by down-regulating plasma low-density lipoprotein cholesterol (LDL-C), triglyceride (TG), and total cholesterol (TC) levels [29,30]. The active phenolic and flavonoid substances in MO exhibit anti-lipid peroxidation effects and act as oxygen radical scavengers to protect the meat from oxidation [31]. The antioxidant properties of *M. oleifera* leaf were also reported to improve goat meat quality (chemical composition, color, and lipid stability) [32].

We previously conducted a pilot study on the liquid fermentation of MO with *Bacillus subtilis* SIX-15. The fermentation products exhibited a 16.98% reduction in tannin level, an 18.75% increase in 1,1-diphenyl-2-picrylhydrazyl (DPPH) scavenging, and a 35.75% increase in total phenol content. Blunt snout bream (*Megalobrama amblycephala*) is a major farmed fish in China. However, this species is highly sensitive to environmental stressors and is vulnerable to disease. Therefore, this research was conducted to study the effects of MO and its fermentation products of FMO on growth performance, feed utilization, and hepatic antioxidant capacity as aquatic animal feed supplements of *M. amblycephala* juveniles.

## 2. Materials and Methods

### 2.1. Ethical Statement

All animal handling procedures were in accordance with the guidelines of the Animal Care Advisory Committee of the Chinese Academy of Fishery Sciences (Authorization No. 20200903001).

### 2.2. Experimental Diets

*M. oleifera* leaves were purchased from Greenway Agriculture Co., Ltd. (Zhongshan, China). MO was obtained by crushing *M. oleifera* leaves and removing solid impurities at 60 mesh. FMO was prepared using *Bacillus subtilis* SIX-15 (Table S1 and Figure S1). Firstly, *Bacillus subtilis* SIX-15 was incubated at 35 °C for 18–20 h in Luria–Bertani broth medium. The fermentation substrate, prepared according to the mass ratio of MO: sterilized water = 1:9. *Bacillus subtilis* SIX-15, was inoculated into sterilized *M. oleifera* leaf mixture to $2 \times 10^5$ cfu/mL and fermented for 48 h at 35 °C. Thereafter, FMO was prepared after being freeze-dried.

Five iso-nitrogenous and iso-energetic diets were designed in this experiment, including the basal diet (control), MO2 diet (2.2% MO was added to the basal diet in place of 1.1% cottonseed meal), MO4 diet (4.4% MO was added to the basal diet in place of 2.2% cottonseed meal), FMO2 diet (2.2% FMO was added to the basal diet in place of 1.1% cottonseed meal), and FMO4 diet (4.4% FMO was added to the basal diet in place of 2.2% cottonseed meal) (Table 1). All ingredients were sieved to remove any solid impurities and then thoroughly mixed. Afterwards, soybean oil and water were gradually added to produce a sinking pellet (2 mm) using an F-26(II) pelletizer (South China University of Technology, Guangzhou, China). Once air-dried, these diets were placed in airtight bags and kept at −20 °C until further use.

**Table 1.** The formulation and nutrient composition of the experimental diets.

| Ingredients (%) | Control | MO2 | MO4 | FMO2 | FMO4 |
|---|---|---|---|---|---|
| MO [1] | | 2.2 | 4.4 | | |
| FMO [2] | | | | 2.2 | 4.4 |
| Fish meal [3] | 4.3 | 4.3 | 4.3 | 4.3 | 4.3 |
| Soybean meal [3] | 21.6 | 21.6 | 21.6 | 21.6 | 21.6 |
| Rapeseed meal [3] | 23.8 | 23.8 | 23.8 | 23.8 | 23.8 |
| Cottonseed meal [3] | 16.4 | 15.3 | 14.2 | 15.3 | 14.2 |
| Wheat meal [3] | 12.2 | 12.2 | 12.2 | 12.2 | 12.2 |
| Rice bran [3] | 10.9 | 12.0 | 13.1 | 12.0 | 13.1 |
| Wheat bran [3] | 4.4 | 2.2 | 0.3 | 2.2 | 0.3 |
| Soybean oil | 2.5 | 2.5 | 2.2 | 2.5 | 2.2 |
| Calcium dihydrogen phosphate [4] | 1.0 | 1.0 | 1.0 | 1.0 | 1.0 |
| Vitamin mix and Mineral mix [5] | 1.0 | 1.0 | 1.0 | 1.0 | 1.0 |
| Vitamin C [4] | 0.1 | 0.1 | 0.1 | 0.1 | 0.1 |
| Choline chloride [4] | 0.4 | 0.4 | 0.4 | 0.4 | 0.4 |
| Microcrystalline cellulose [4] | 1.0 | 1.0 | 1.0 | 1.0 | 1.0 |
| Bentonite [4] | 0.4 | 0.4 | 0.4 | 0.4 | 0.4 |
| Nutrient composition (determination of dry basis) | | | | | |
| Crude protein % | 35.71 | 35.18 | 35.73 | 35.84 | 35.91 |
| Crude lipid % | 10.42 | 10.43 | 10.23 | 10.75 | 10.79 |
| Gross energy MJ·kg$^{-1}$ | 19.87 | 19.96 | 19.63 | 19.38 | 19.66 |

[1] Provided by Greenway Agriculture Co., Ltd. (Zhongshan, China). [2] Provided by Freshwater Fisheries Research Center of Chinese Academy of Fishery Sciences. (Wuxi, China). [3] Provided by Tongwei Co., Ltd. (Wuxi, China). [4] Provided by Wuxi Hanove Animal Health Products Co., Ltd. (Wuxi, China). [5] Vitamin and mineral mix provided by Wuxi Hanove Animal Health Products Co., Ltd. (Wuxi, China). Vitamin mix (mg/kg) = 10 mg vitamin B1, 20 mg vitamin B2, 30 mg vitamin B6, 0.02 mg vitamin B12, 5 mg folic acid, 50 mg calcium pantothenate, 100 mg inositol, 100 mg niacin, 0.5 mg vitamin H, 100 mg vitamin C, 110 mg vitamin A, 20 mg vitamin D, 100 mg vitamin E, and 10 mg vitamin K. Mineral mix (g/kg or mg/kg) = 15 g $MgSO_4$, 2.5 g $FeSO_4$, 31 mg $CuSO_4$, 162 mg $MnSO_4$, 353 mg $ZnSO_4$, 3 mg $KIO_3$, 3 mg $Na_2SeO_3$, and 1 mg $CoSO_4$.

### 2.3. Experimental Fish

The feeding trial was conducted at the Freshwater Fisheries Research Center (FFRC) of the Chinese Academy of Fishery Sciences (120.92 E, 31.43 N). *M. amblycephala* 'Huahai No.1' were obtained from the National *M. amblycephala* stock farm (Wuhan, China). Healthy and similarly sized *M. amblycephala* juveniles (initial body weight of $20.27 \pm 0.11$ g; 300 individuals) were randomly assigned to 15 floating cages (1 m $\times$ 1 m $\times$ 1 m, 20 fish per floating cage). *M. amblycephala* juveniles were fed with commercial feed containing 33.0% protein and 7.0% lipid (Tongwei Co., Ltd., Wuxi, China) and were allowed to acclimate to the farming environment for one week. The control, MO2, MO4, FMO2, and FMO4 diets were randomly assigned to three floating cages each. The fish were hand-fed carefully three times daily at 7:30, 11:30, and 17:30 until apparent satiation (based on visual observation) for 56 days. During the feeding trial, the water temperature ranged from 23 °C to 27 °C. The dissolved oxygen, ammonia nitrogen, and pH were $\geq 6.0$ mg/L, $0.029 \pm 0.002$ mg/L, and $7.2 \pm 0.2$, respectively.

### 2.4. Sample Collection

At the end of the feeding trial, all the fish were fasted for 24 h, anesthetized with 100 mg/L of tricaine methanesulfonate (MS-222), and weighed to calculate growth performance in terms of weight gain rate (WGR), specific growth rate (SGR), feed coefficient rate (FCR), protein efficiency ratio (PER), and condition factor (CF).

Four fish were then randomly collected from each cage for sampling. Blood of four fish was obtained using disposable medical syringes from the caudal vein. The supernatant plasma was collected after centrifuging at 4 °C and 4000 r/min for 10 min. The plasma was kept at $-20$ °C until biochemical parameters were determined. Three fish were randomly selected from each cage and immediately dissected to collect liver tissue and dorsal muscle. Liver samples were divided into two parts, one part stored at $-20$ °C for antioxidant parameters assay and the other stored at $-80$ °C for genes relative expressions assay. Dorsal muscle samples from all the groups were stored at $-20$ °C for muscle composition determination. Dorsal muscle samples from the control, MO4, and FMO4 groups were collected and stored at 4 °C for determining the free amino acid contents.

### 2.5. Laboratory Analysis

2.5.1. Growth Performance

Weight gain rate (WGR, %) = 100 $\times$ (final body weight (g) $-$ initial body weight (g))/initial body weight (g).

Specific growth rate (SGR, %/day) = 100 $\times$ [ln (final body weight (g)) $-$ ln (initial body weight (g))]/feeding days.

Feed coefficient rate (FCR) = dry feed intake (g)/wet weight gain (g).

Protein efficiency ratio (PER, %) = 100 $\times$ wet weight gain (g)/feed protein intake (g).

Condition factor (CF, g/cm$^3$) = 100 $\times$ final body weight (g)/body length (cm)$^3$.

2.5.2. Determination of Feed and Muscle Composition

Feed and muscle nutrient contents were examined according to the AOAC [33] criteria. Briefly, the samples were dried with hot air (105 °C), after which the moisture contents were calculated. Crude protein and crude lipid contents were determined via the Kjeldahl and Soxhlet methods, respectively. Ash contents were measured after burning the samples at 550 °C for 5 h. Gross energy contents were measured with an oxygen bomb calorimeter (IKA C6000, IKA, Staufen, Germany).

The free amino acid contents in muscle were examined based on the method described by Zeng et al. [34]. The samples were analyzed through high-performance liquid chromatography (AG1100, Agilent Technologies Co., Ltd., Santa Clara, CA, USA). Mobile phase A consisted of 0.8% (*m*/*v*) sodium acetate and 0.0225% (*v*/*v*) triethylamine solution. Mobile phase B was a mixture of 2% (*m*/*v*) sodium acetate buffer (pH 7.2), acetonitrile, and methanol at a 1:2:2 (*v*/*v*) ratio.

### 2.5.3. Determination of Plasma Biochemical Parameters

Triglycerides (TG), glucose (GLU), total cholesterol (TC), and low-density lipoprotein cholesterol (LDL-C) were determined on a Mindray BS-400 automated biochemistry analyzer (Mindray Bio-Medical Electronics Co., Ltd., Shenzhen, China) using commercial kits purchased from Zhicheng Bio-Technology Co., Ltd. (Shanghai, China).

### 2.5.4. Determination of Hepatic Antioxidant Parameters

The liver samples were homogenized in pre-cooled normal saline according to the protocol. The samples were then centrifuged (3500 r/min, 10 min) and the supernatant was collected to measure the liver antioxidant parameters. Catalase (CAT), total superoxide dismutase (T-SOD), glutathione peroxidase (GPX) activities, and glutathione (GSH), malondialdehyde (MDA) levels were measured by the corresponding commercial kits (Jiancheng Bioengineering Institute, Nanjing, China).

### 2.5.5. Real-Time PCR (qRT-PCR) Analysis on Genes Relative Expressions

Total RNA was extracted from fish liver using trizol (TaKaRa Biomedical Technology Co., Ltd., Dalian, China). The concentration and purity of the total RNA were quantified using Nanodrop 2000 (Thermo Fisher Scientific Inc., Waltham, MA, USA). The mRNA expression levels of toll-like receptor 4 (*tlr4*), nuclear factor-kappa B (*nf-κb*), tumor protein P53 (*p53*), interleukin 8 (*il-8*), glucose-6-phosphatase (*g6pase*), lipoprotein lipase (*lpl*), pyruvate kinase (*pk*), peroxisome proliferator activated receptor-β (*ppar-β*), and acetyl-CoA carboxylase α (*acc-α*) were determined by qRT-PCR using the TB Green™ Premix Ex Taq™ II (TaKaRa Biomedical Technology Co., Ltd., Dalian, China) on the CFX96 instrument (Bio-Rad Laboratories, Inc., Hercules, CA, USA). beta-cytoskeletal actin (*β-actin*) was taken as the reference gene. The sequences of the gene-specific primers were designed by the national center for biotechnology information (NCBI), and the primers were synthesized by Sangon Biotech Co., Ltd. (Shanghai, China) (Table 2). The qRT-PCR procedure was as follows: pre-denaturing at 95 °C for 30 s; 39 cycles of denaturation for 5 s at 95 °C and 30 s at 60 °C (annealing temperature); and extension at 95 °C for 10 s. The relative expressions of the target genes were calculated using the $2^{-\Delta\Delta Ct}$ method.

**Table 2.** Primer sequences for qRT-PCR.

| Genes | | Primer Sequence (5′–3′) | Accession No. | Product Length (bps) |
|---|---|---|---|---|
| *tlr4* | Forward | TAATGGGCAGCCGTAAAGTC | XM_048204247.1 | 114 |
| | Reverse | TGGCATTGCGTTCCATAATA | | |
| *nf-κb* | Forward | AGTCCGATCCATCCGCACTA | XM_048176853.1 | 85 |
| | Reverse | ACTGGAGCCGGTCATTTCAG | | |
| *il-8* | Forward | CAGAGAGTCGACGCATTGGT | XM_048197357.1 | 184 |
| | Reverse | ATTCACGGTGCTTTGTTGGC | | |
| *p53* | Forward | CCATCCTCACAATCATCAC | XM_048187452.1 | 114 |
| | Reverse | TGCTCTCCTCAGTTTTCCT | | |
| *g6pase* | Forward | TTCAGTGTCACGCTGTTCCT | XM_048171060.1 | 119 |
| | Reverse | TCTGGACTGACGCACCATTT | | |
| *pk* | Forward | GCCGAGAAAGTCTTCATCGCACAG | XM_048152870.1 | 157 |
| | Reverse | CGTCCAGAACCGCATTAGCCAC | | |
| *ppar-β* | Forward | CATCCTCACGGGCAAGAC | XM_048209548.1 | 153 |
| | Reverse | CACTGGCAGCGGTAGAAG | | |
| *acc-α* | Forward | TCTGCCCTCTATCTGTCT | XM_048189972.1 | 162 |
| | Reverse | ATGCCAATCTCATTTCCT | | |
| *lpl* | Forward | GCCACGAGTGTTGGTGTGAA | XM_048164066.1 | 91 |
| | Reverse | TGGCCCTAGCTTTGAGTACG | | |
| *β-actin* | Forward | TCGTCCACCGCAAATGCTTCTA | AY170122.2 | 190 |
| | Reverse | CCGTCACCTTCACCGTTCCAGT | | |

*tlr4*: toll-like receptor 4; *nf-κb*: nuclear factor-kappa B; *il-8*: interleukin 8; *p53*: tumor protein P53; *g6pase*: glucose-6-phosphatase; *lpl*: lipoprotein lipase; *pk*: pyruvate kinase; *ppar-β*: peroxisome proliferator activated receptor-β; *acc-α*: acetyl-CoA carboxylase α; and *β-actin*: beta-cytoskeletal actin.

*2.6. Statistical Analysis*

One-way ANOVA was performed using SPSS 22.0 software (SPSS Inc., Chicago, IL, USA) after testing normality and homogeneity of variance. A Tukey's test ($p < 0.05$) was used to detect differences in means among experimental groups. The data are reported as means $\pm$ standard error. A $p$ value $< 0.05$ was considered statistically significant. All figures were generated using GraphPad Prism 7.0.

## 3. Results

*3.1. Growth Performance*

As shown in Table 3, FBW, WGR, SGR, FCR, and PER were not significantly affected by experimental diets ($p > 0.05$). CF was significantly lower in the FMO4 groups compared to the control group ($p < 0.05$).

**Table 3.** Growth performance of *M. amblycephala* juveniles fed with the experimental diets.

| Parameters | IBW (g) | FBW (g) | WGR (%) | SGR (% day$^{-1}$) | FCR | PER (%) | CF (g/cm$^3$) |
|---|---|---|---|---|---|---|---|
| Control | 20.37 $\pm$ 0.07 | 48.60 $\pm$ 0.69 | 138.56 $\pm$ 3.08 | 1.64 $\pm$ 0.03 | 1.51 $\pm$ 0.03 | 1.32 $\pm$ 0.04 | 2.27 $\pm$ 0.06 [b] |
| MO2 | 20.15 $\pm$ 0.11 | 43.26 $\pm$ 1.70 | 114.74 $\pm$ 9.19 | 1.44 $\pm$ 0.08 | 1.79 $\pm$ 0.08 | 1.40 $\pm$ 0.01 | 2.14 $\pm$ 0.05 [ab] |
| FMO2 | 20.25 $\pm$ 0.07 | 45.06 $\pm$ 2.33 | 126.46 $\pm$ 14.11 | 1.53 $\pm$ 0.12 | 1.59 $\pm$ 0.18 | 1.60 $\pm$ 0.10 | 2.10 $\pm$ 0.05 [ab] |
| MO4 | 20.20 $\pm$ 0.15 | 46.12 $\pm$ 2.20 | 128.33 $\pm$ 10.51 | 1.55 $\pm$ 0.09 | 1.58 $\pm$ 0.12 | 1.66 $\pm$ 0.13 | 2.12 $\pm$ 0.06 [ab] |
| FMO4 | 20.36 $\pm$ 0.16 | 47.10 $\pm$ 0.53 | 131.42 $\pm$ 3.76 | 1.58 $\pm$ 0.03 | 1.54 $\pm$ 0.09 | 1.63 $\pm$ 0.26 | 2.05 $\pm$ 0.02 [a] |
| $p$ value | 0.267 | 0.310 | 0.498 | 0.499 | 0.479 | 0.369 | 0.042 |

Values in the same column with different superscripts are significantly different ($p < 0.05$, Tukey's test). IBW: initial body weight; FBW: final body weight; WGR: weight gain rate; SGR: specific growth rate; FCR: feed coefficient rate; PER: protein efficiency ratio; and CF: condition factor.

*3.2. Muscle Nutrient Composition*

The nutrient concentration of the muscle is summarized in Table 4. The moisture contents of the FMO2 and MO4 groups were prominently higher than that of the MO2 and FMO4 groups ($p < 0.05$). The MO4 and FMO4 groups increased muscle crude protein contents compared with the MO2 and FMO2 groups ($p < 0.05$). Moreover, the crude protein content was prominently reduced in the FMO2 group compared to the control group ($p < 0.05$). A significantly higher content of muscle crude lipid was detected in the FMO4 group versus the MO2 group ($p < 0.05$). Ash content was not affected by the experimental diets.

**Table 4.** Composition of muscles in *M. amblycephala* juveniles fed with the experimental diets.

| Muscle Composition | Moisture % | Crude Protein % | Crude Lipid % | Ash % |
|---|---|---|---|---|
| Control | 77.89 $\pm$ 0.10 [ab] | 19.52 $\pm$ 0.12 [bc] | 2.18 $\pm$ 0.14 [ab] | 1.27 $\pm$ 0.02 |
| MO2 | 77.62 $\pm$ 0.21 [a] | 19.10 $\pm$ 0.11 [ab] | 1.99 $\pm$ 0.07 [a] | 1.29 $\pm$ 0.03 |
| FMO2 | 78.65 $\pm$ 0.10 [b] | 18.89 $\pm$ 0.15 [a] | 2.16 $\pm$ 0.14 [ab] | 1.28 $\pm$ 0.01 |
| MO4 | 78.46 $\pm$ 0.21 [b] | 19.77 $\pm$ 0.11 [c] | 2.15 $\pm$ 0.11 [ab] | 1.31 $\pm$ 0.05 |
| FMO4 | 77.34 $\pm$ 0.27 [a] | 19.71 $\pm$ 0.09 [c] | 2.60 $\pm$ 0.07 [b] | 1.25 $\pm$ 0.03 |
| $p$ value | <0.001 | <0.001 | 0.008 | 0.812 |

Values in the same column with different superscripts are significantly different ($p < 0.05$, Tukey's test).

The free amino acid profiles in muscle of *M. amblycephala* juveniles fed with the control, MO4, and FMO4 diets were also analyzed (Table 5). Specifically, the three amino acids with the highest levels in muscle were histidine, glycine, and threonine. Based on their taste properties, the levels of free amino acids exhibited the following order: the total amount of bitter amino acids (TBAA) > the total amount of sweet amino acids (TSAA) > the total amount of umami amino acids (TUAA). Moreover, the MO4 group exhibited significant increases in the total amount of TSAA compared with the FMO4 group ($p < 0.05$). Specifically, a remarkable improvement in glycine content was observed in the

MO4 group compared to the control and the FMO4 groups ($p < 0.05$). Moreover, the MO4 and FMO4 groups exhibited substantially lower alanine and threonine contents compared to the control group ($p < 0.05$). The muscle proline content was also significantly enhanced in the FMO4 group compared to the MO4 group ($p < 0.05$).

**Table 5.** Free amino acid composition of muscles in *M. amblycephala* juveniles fed with the experimental diets.

| Content mg/100 g | Control | MO4 | FMO4 | *p* Value |
|---|---|---|---|---|
| Aspartic acid | 3.31 ± 0.05 [b] | 3.07 ± 0.04 [a] | 3.08 ± 0.05 [a] | 0.018 |
| Glutamic acid | 7.48 ± 0.42 | 7.54 ± 0.39 | 6.75 ± 0.04 | 0.166 |
| Total umami amino acids (TUAA) | 10.79 ± 0.15 | 10.61 ± 0.43 | 9.83 ± 0.15 | 0.094 |
| Threonine | 31.28 ± 0.32 [b] | 28.78 ± 0.53 [a] | 27.65 ± 0.14 [a] | 0.048 |
| Glycine | 105.30 ± 1.32 [a] | 122.77 ± 0.79 [b] | 105.51 ± 0.21 [a] | 0.006 |
| Alanine | 27.56 ± 0.23 [b] | 22.97 ± 0.68 [a] | 23.43 ± 0.02 [a] | 0.004 |
| Proline | 14.65 ± 0.34 [ab] | 9.55 ± 1.13 [a] | 17.57 ± 2.65 [b] | 0.038 |
| Serine | 1.88 ± 0.72 | 3.06 ± 0.18 | 2.41 ± 0.21 | 0.124 |
| Total sweetish amino acids (TSAA) | 180.67 ± 0.85 [ab] | 187.13 ± 1.75 [b] | 176.57 ± 2.83 [a] | 0.018 |
| Valine | 4.06 ± 0.06 [b] | 3.60 ± 0.09 [a] | 3.91 ± 0.13 [ab] | 0.037 |
| Methionine | 2.06 ± 0.04 [a] | 2.64 ± 0.17 [b] | 2.80 ± 0.03 [b] | 0.005 |
| Isoleucine | 2.40 ± 0.08 | 2.09 ± 0.13 | 2.48 ± 0.03 | 0.057 |
| Leucine | 4.90 ± 0.05 [b] | 3.83 ± 0.22 [a] | 4.33 ± 0.08 [ab] | 0.005 |
| Phenylalanine | 2.09 ± 0.36 [a] | 3.44 ± 0.78 [ab] | 5.05 ± 0.07 [b] | 0.007 |
| Histidine | 555.23 ± 6.91 | 569.57 ± 13.25 | 584.75 ± 7.99 | 0.184 |
| Lysine | 20.64 ± 0.24 [b] | 11.82 ± 1.18 [a] | 17.59 ± 0.22 [b] | <0.001 |
| Argnine | 7.76 ± 0.07 [c] | 4.78 ± 0.39 [a] | 6.67 ± 0.02 [b] | <0.001 |
| Tyrosine | 4.50 ± 0.10 [a] | 5.74 ± 0.43 [b] | 4.88 ± 0.21 [ab] | 0.025 |
| Cysteine | 11.04 ± 0.36 [b] | 8.68 ± 0.66 [b] | 5.32 ± 1.03 [a] | 0.004 |
| Total bitter amino acids (TBAA) | 614.68 ± 6.48 | 616.20 ± 14.60 | 637.78 ± 7.89 | 0.142 |
| Total free amino acids (TFAA) | 807.06 ± 7.69 [a] | 813.94 ± 15.53 [a] | 824.18 ± 5.12 [b] | 0.029 |

The data represent the mean ± SEM. Means in the same row with different superscripts are significantly different ($p < 0.05$, Tukey's test). Total umami amino acids (TUAA): aspartic acid + glutamic acid; total sweetish amino acids (TSAA): threonine + glycine + alanine + proline + serine; total bitter amino acids (TBAA): valine + methionine + isoleucine + leucine + phenylalanine + histidine + lysine + arginine + tyrosine + cysteine.

## 3.3. Plasma Biochemical Indices

The plasma biochemical indices are illustrated in Figure 1. The FMO2 and FMO4 groups exhibited significantly higher GLU levels compared to the control group ($p < 0.05$). A further noticeable increase in GLU content was observed in the FMO4 group compared to the MO2 and MO4 groups ($p < 0.05$). The TG content was significantly increased in the FMO2 group versus the control group ($p < 0.05$). Moreover, a distinct increment in TG content was observed in the FMO2 and FMO4 groups relative to the MO4 group ($p < 0.05$). Furthermore, the content of LDL-C was significantly increased in the FMO4 group relative to the MO2 group ($p < 0.05$).

## 3.4. Hepatic Antioxidant Parameters

The hepatic antioxidant capacities of *M. amblycephala* juveniles are illustrated in Figure 2. No significant differences were detected in the levels of MDA and GPX ($p > 0.05$). CAT activity was significantly increased in the FMO4 group versus the control group ($p < 0.05$). The MO4, FMO2, and FMO4 groups exhibited a distinct enhancement in T-SOD activities compared to the control group ($p < 0.05$). Furthermore, the T-SOD activity of the FMO4 group was significantly higher than that of the MO2 group ($p < 0.05$). The FMO2 group exhibited a notable increase in liver GSH content compared to the MO2 and FMO4 groups ($p < 0.05$).

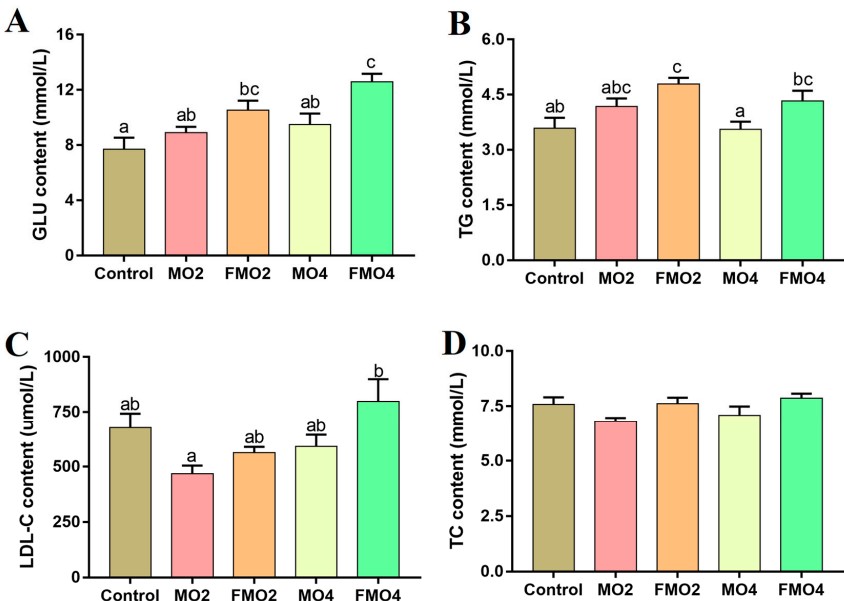

**Figure 1.** Effects of dietary MO and FMO on plasma biochemical parameters: GLU: glucose (**A**); TG: triglycerides (**B**); LDL-C: low-density lipoprotein cholesterol (**C**); and TC: total cholesterol (**D**). Values with different alphabetical superscripts above bars are significantly different ($p < 0.05$, Tukey's test).

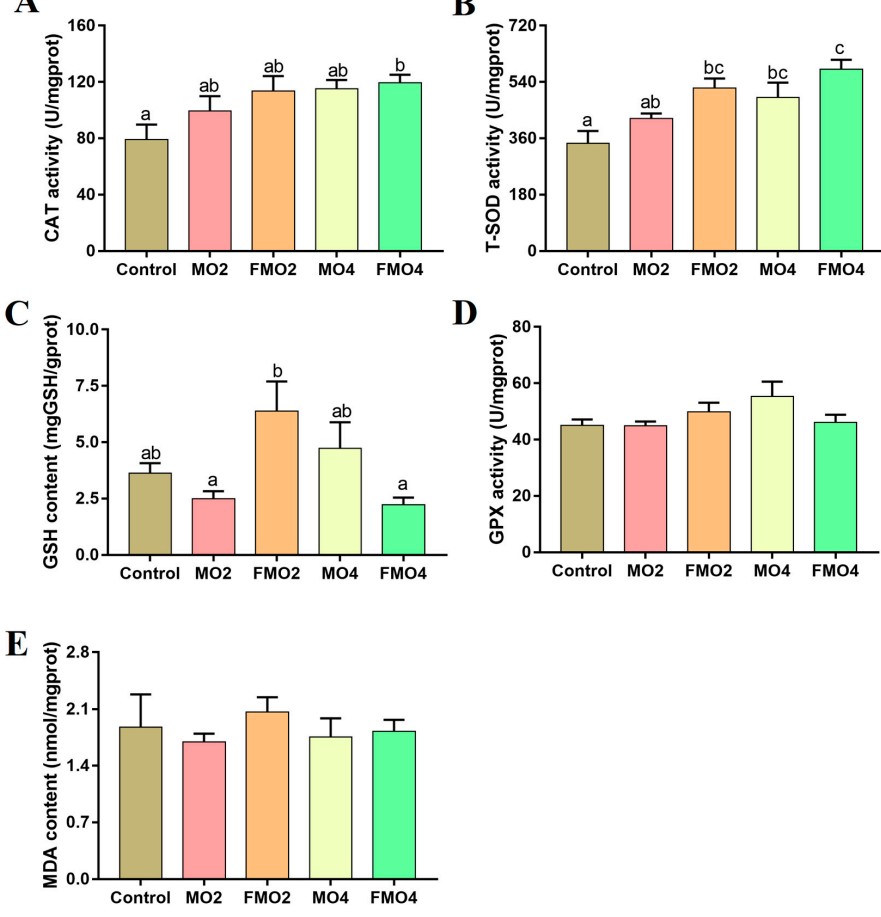

**Figure 2.** Effects of dietary MO and FMO on hepatic CAT: catalase (**A**); T-SOD: total superoxide dismutase (**B**); GSH: glutathione (**C**); GPX: glutathione peroxidase (**D**); and MDA: malondialdehyde (**E**) levels. Values with different alphabetical superscripts above bars are significantly different ($p < 0.05$, Tukey's test).

### 3.5. Gene Expressions Related to Glucose and Lipid Metabolism in the Liver

Figure 3 illustrates the mRNA expression of target genes related to hepatic glucose and lipid metabolism. The hepatic mRNA expressions of *g6pase* in the MO2, MO4, FMO2, and FMO4 groups were considerably higher than those of the control group ($p < 0.05$). Furthermore, a noticeable reduction in *pk* mRNA expression was observed in the FMO2 and FMO4 groups compared to the control and MO2 groups ($p < 0.05$). The MO4 group exhibited a remarkable reduction in *acc-α* mRNA expression levels relative to the FMO2 and FMO4 groups ($p < 0.05$). The *lpl* mRNA levels of the MO4, FMO2, and FMO4 groups were down-regulated relative to the MO2 group ($p < 0.05$).

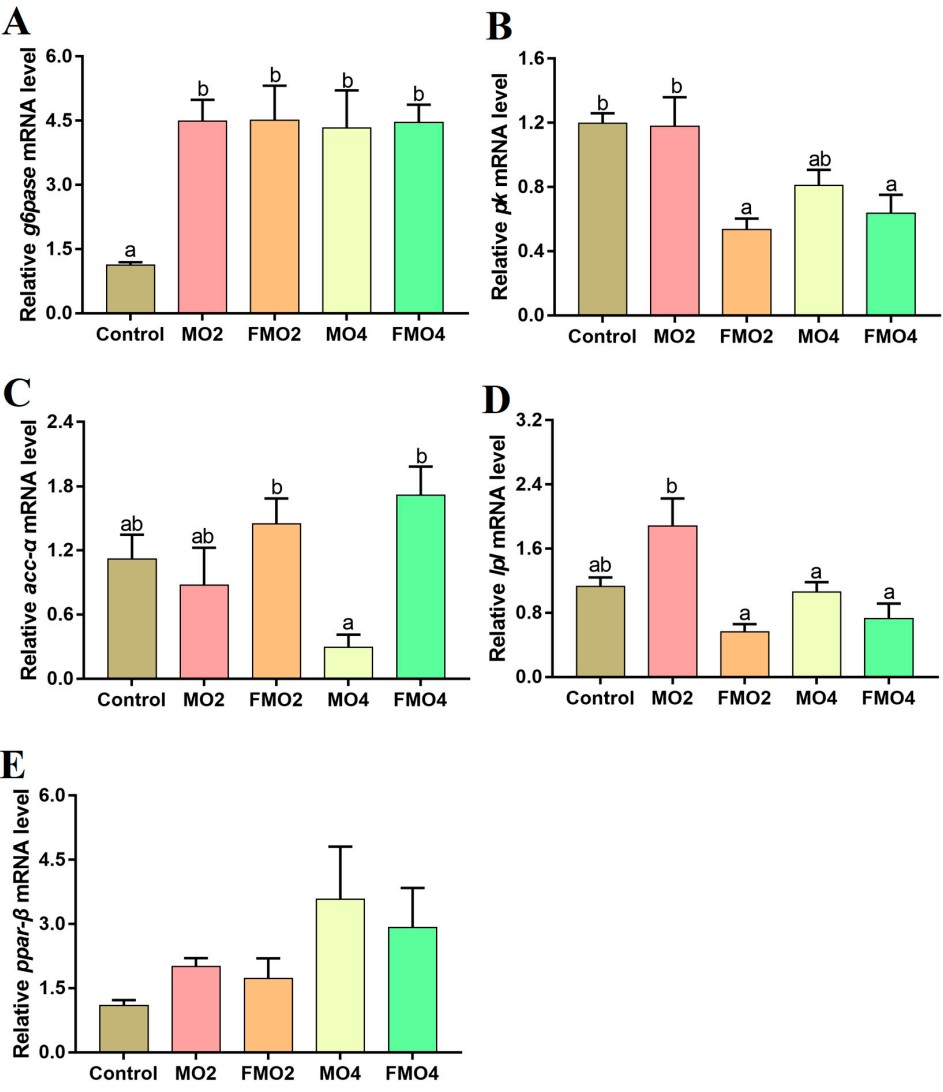

**Figure 3.** Effects of dietary MO and FMO on hepatic *g6pase*: glucose-6-phosphatase (**A**); *pk*: pyruvate kinase (**B**); *acc-α*: acetyl-CoA carboxylase α (**C**); *lpl*: lipoprotein lipase (**D**); and *ppar-β*: peroxisome proliferator activated receptor-β (**E**) expression levels. Values with different alphabetical superscripts above bars are significantly different ($p < 0.05$, Tukey's test).

### 3.6. Gene mRNA Expressions Related to the Inflammatory Response in the Liver

Figure 4 illustrates the mRNA expressions of genes related to inflammatory responses in the liver. The MO2, MO4, FMO2, and FMO4 groups showed a noticeable reduction in *tlr4* mRNA expression levels versus the control group ($p < 0.05$). A noticeable reduction in *nf-κb* mRNA expression level was observed in the MO4 group compared to the MO2 group ($p < 0.05$). Furthermore, the *nf-κb* mRNA expression levels of FMO2 and FMO4 groups were significantly lower than the control and MO2 groups ($p < 0.05$). The FMO2 group

exhibited a remarkable reduction in *il-8* mRNA expression level relative to the control and MO2 groups ($p < 0.05$). Furthermore, the *il-8* mRNA expression level was significantly reduced in the MO4 group relative to the MO2 group ($p < 0.05$). A distinct decrease in the *p53* expression level was observed in the FMO2 and FMO4 groups compared to the MO2 group ($p < 0.05$).

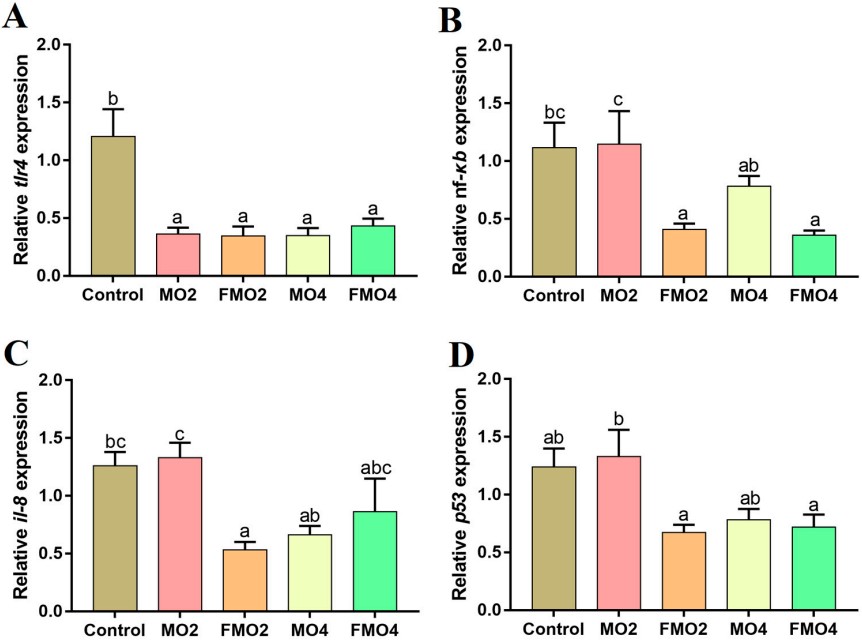

**Figure 4.** Effects of dietary MO and FMO on hepatic *tlr4*: toll-like receptor 4 (**A**); *nf-κb*: nuclear factor-kappa B (**B**); *il-8*: interleukin 8 (**C**); and *p53* (**D**) expression levels. Values with different alphabetical superscripts above bars are significantly different ($p < 0.05$, Tukey's test).

## 4. Discussion

MO has been demonstrated to enhance the growth performance of common carp (*Cyprinus carpio*) [35] and *O. niloticus* [28], and FMO has been reported to increase the growth performance of *C. gibelio* juveniles [9]. In this study, supplementation with MO and FMO had no negative effects on the growth performance of *M. amblycephala* juveniles, which was consistent with previous studies on guppy (*Poecilia reticulata*) [36] and *O. niloticus* [37]. These discrepancies may be attributed to differences in fish size and feeding conditions, as well as fish species [28]. Our results indicated that the supplementation of FMO appeared to affect the body shape of *M. amblycephala* juveniles as reflected in the decreased CF. Previous studies have revealed that supplementation of fermentation products such as protein ingredients reduces CF in largemouth bass (*Micropterus salmoides*) [38].

Studies have found that replacing 10% soybean meal or fish meal with MO increased the crude protein contents of whole fish such as rohu (*Labeo rohita*) [39] and *O. niloticus* [40]. Another study reported that phytase-supplemented MO improved the nutrient profiles (protein and lipid) of *Catla catla* (Hamilton, 1822) [41]. These results were also reflected in our feeding trial, where the supplementation of 4.4% MO and 4.4% FMO significantly enhanced the muscle crude protein contents of *M. amblycephala* juveniles. Studies have shown that prolonged consumption of formulated diets containing high levels of plant protein and plant oil results in increased muscle hardness, reduced sweetness, and weakened odor intensity in fish [42]. Threonine, glycine, alanine, proline, and serine are classified as sweet amino acids [43]. Similarly, previous studies reported that supplementing non-fishmeal diets with (2-carboxyethyl) dimethylsulfonium bromide [44] and guanidinoacetic acid [45] significantly increased the contents of sweet amino acids in grass carp (*Ctenopharyngodon idella*). Moreover, substituting 30% of fish meal with MO had no significant effects on the texture and flavor profile of *C. gariepinus* meat compared to the control [46]. In the

present study, we found that dietary supplementation with 4.4% MO increased the total content of sweet amino acids in muscle, thereby probably improving the meat flavor of *M. amblycephala* juveniles.

Lipid accumulation is a complex process involving the ingestion, transportation, and catabolism of lipids. The catabolism of triacylglycerols is also a crucial process influencing lipid deposition in particular tissues. Reduced hepatic lipid accumulation and TG catabolism were observed in large yellow croaker (*Larimichthys crocea*) fed with a low-lipid diet [47]. Intracellular lipid accumulation in muscle and liver reflects a phenomenon of ectopic TG distribution in locations other than adipose tissue, which results from an impairment of the cellular mechanisms that regulate lipid storage and utilization [48]. Studies have found that increased levels of LDL-C, TG, and GLU in the blood promoted lipid deposition in the muscle of golden pompano (*Trachinotus ovatus*) and pacific abalone (*Haliotis discus hannai*) [49,50]. In our study, the plasma LDL-C, TG, and GLU levels and muscle lipid content were markedly increased in the group fed with the diet containing 4.4% FMO. We speculated that FMO promoted lipogenesis in the present study. This was consistent with the results of a previous study, in which *C. gariepinus* fed with a diet containing 20% MO exhibited improvements in muscular lipid deposition [51]. Previous studies have also reported that probiotics may improve the synthesis of vitamin B12, biotin, and fatty acids, among other nutrients [52,53], thereby positively influencing the wellness of organisms. Additionally, we analyzed the expression of target genes related to lipogenesis and glucose metabolism. Our findings indicated that FMO supplementation up-regulated hepatic *acc-α* mRNA expression and down-regulated *lpl* mRNA expression. Acc-α is a key enzyme involved in fatty acid biosynthesis and can regulate lipogenesis by participating in the synthesis of long-chain fatty acids and β-oxidation [54]. Moreover, lpl acts as a rate-limiting enzyme within the lipid hydrolysis process and contributes to the absorption of lipids from food and plasma lipoprotein metabolism [55]. Glucose metabolism can preserve the energy homeostasis of vital functions in animals and has a crucial role in modulating glucose and lipid metabolism [56]. A 15 mM dose of glucose was found to induce TG accretion in the plasma of yellow catfish (*Pelteobagrus fulvidraco*) [57]. Elevated plasma GLU level increases the expression levels of lipogenic genes such as *acc-α* and fat synthase (*fas*), which in turn enhances lipid deposition [58,59]. Furthermore, increased plasma GLU level inhibits the transcription of peroxisome proliferator-activated receptors-α (*ppar-α*) and carnitine palmitoyltransferase 2 (*cpt-2*), thereby inhibiting fatty acid oxidation [60]. G6pase transforms glucose-6-phosphate to free glucose in the last step of the gluconeogenesis pathway [61]. The process of glycogen utilization can be completed by glycolysis, and pk acts as the rate-limiting enzyme in the last step of glycolysis [62]. In this study, supplementation with FMO increased plasma GLU levels and provided energy to the juvenile fish [63] by enhancing the expression of *g6pase* mRNA and down-regulating the *pk* mRNA expression level, thus stimulating hepatic gluconeogenesis and decreasing the glycolytic capacity of the liver. Therefore, we speculated that 4.4% FMO promoted lipid accumulation in *M. amblycephala* juveniles due to an increase in plasma GLU level, which reduced *lpl* and increased *acc-α* mRNA expression levels, thereby promoting lipid accumulation in locations other than adipose tissue.

*M. oleifera* leaves are rich in vitamin C, vitamin A, and phenolic compounds such as quercetin and flavonoids. Studies have shown that *M. oleifera* leaf extracts increased antioxidant activity and slowed down oxidative damage in goats, rats, and other mammals [64]. In aquatic animals, dietary administration of MO and its extracts enhanced the tissue antioxidant activities of Nile tilapia and *M. rosenbergii* [13,19,20]. Moreover, dietary supplementation with FMO increased the plasma SOD, CAT, and GPX activities of Gibel carp [9]. Fermentation changes the type and volume of active ingredients in the substrate, and the resulting small molecule glycosides have stronger antioxidant activities [9]. In this study, dietary supplementation with 4.4% FMO increased the hepatic CAT and T-SOD activities. The primary role of T-SOD is to scavenge intracellular superoxide anions and produce non-toxic oxygen as well as less harmful hydrogen peroxide [13,16].

CAT effectively scavenges intracellular hydrogen peroxide and catalyzes the breakdown of hydrogen peroxide to produce water and oxygen [19,20]. In other words, 4.4% FMO improved the hepatic antioxidant activities of *M. amblycephala* juveniles. The GSH content of the FMO2 group was significantly higher compared to the MO2 and FMO4 groups. GSH scavenges reactive oxygen species (ROS) to alleviate oxidative stress directly [65]. The present results indicated that FMO improved the hepatic antioxidant capacity of *M. amblycephala* juveniles, which was stronger in the dietary 4.4% FMO. Non-specific immunity is the first line of defense against invading pathogens. Toll-like receptors (tlrs) serve as the primary sensors that induce innate immune responses in fish by detecting various microbial components [66]. Among these, tlr4 is a pattern-recognition receptor that recognizes pathogenic microbes and is expressed in the cell membrane [67]. Nf-κb serves as the critical downstream activator of the tlr4 pathway, and therefore *tlr4* activation is ultimately manifested by *nf-κb* activation [68]. Recent studies demonstrated that plant polysaccharides inhibit the expressions of *tlr2*, *nf-κb*, and their downstream inflammatory genes in tissues of *O. niloticus* and Jian carp (*Cyprinus carpio var*. Jian) [69,70]. In this experiment, the diets containing 2.2% and 4.4% FMO down-regulated the mRNA expressions of pro-inflammatory factors of *tlr4* and *nf-κb*, as well as the apoptosis factor *p53*. Our finding was consistent with those of a previous study in which FMO modulated the inflammatory reaction of Gibel carp and improved their tolerance to pathogenic bacteria by modulating the tlr2 signaling pathway [9]. Moreover, *M. oleifera* ethanolic extracts decreased the mRNA expressions of pro-inflammatory factors, thereby attenuating the kidney damage induced by tilmicosin [64]. These findings suggested that dietary 2.2% and 4.4% FMO improved the innate immunity of *M. amblycephala* juveniles by inhibiting the levels of pro-inflammatory factors.

## 5. Conclusions

Dietary supplementation with 2.2% and 4.4% of *M. oleifera* leaf meal or fermented *M. oleifera* leaf meal did not affect the growth performance of *M. amblycephala* juveniles, while dietary 2.2% and 4.4% fermented *M. oleifera* leaf meal enhanced anti-inflammatory capacity of the liver. Dietary 4.4% fermented *M. oleifera* leaf meal supplementation was recommended because it increased the muscle nutrient composition and hepatic antioxidant and anti-inflammatory capacities of the *M. amblycephala* juveniles. Based on the results of our present study, probiotics fermentation improved the physiological status of *M. amblycephala* juveniles fed *M. oleifera* leaf meal. It is necessary to conduct further investigations into the mechanisms by which fermented protein sources enhance nutrient deposition in aquatic animals.

**Supplementary Materials:** The following supporting information can be downloaded at: https://www.mdpi.com/article/10.3390/fermentation9060577/s1, Table S1. The nutrients profile in MO and FMO; Figure S1. Molecular weight distribution of peptides in MO and FMO; * indicates a significant difference ($p < 0.05$; independent *t*-test); ** indicates an extremely significant difference ($p < 0.01$; independent *t*-test).

**Author Contributions:** Conceptualization, Y.L.; data curation, W.J.; formal analysis, Y.Y.; investigation, L.Q.; methodology, Y.Y. and L.M.; project administration, H.S. and X.G.; resources, Y.Z. and Y.L.; software, L.Q.; supervision, X.G. and L.M.; validation, H.S.; writing—original draft, W.J.; writing—review and editing, L.M. All authors have read and agreed to the published version of the manuscript.

**Funding:** This work was supported by China Agriculture Research System (CARS-45) and Science and Technology Innovation Team (Grant No. 2020TD59).

**Institutional Review Board Statement:** All animal handling procedures were in accordance with the guidelines of the Animal Care Advisory Committee of the Chinese Academy of Fishery Sciences (Authorization No. 20200903001).

**Informed Consent Statement:** Not applicable.

**Data Availability Statement:** The authors confirm that the data supporting the findings of this study are available within the manuscript and tables.

**Acknowledgments:** We would like to thank the postgraduate students of the Fish Disease and Nutrition Department, the Freshwater Fisheries Research Center (FFRC), and the Chinese Academy of Fishery Sciences (CAFS) for their help throughout the research period.

**Conflicts of Interest:** The authors declare no conflict of interest.

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
