# Peer review of "The Application of Moringa oleifera Leaf Meal and Its Fermentation Products in the Diet of Megalobrama amblycephala Juveniles"

_fermentation, doi:10.3390/fermentation9060577_

Round 1

Reviewer 1 Report

The application of Moringa oleifera leaf meal and its fermenta-2 tion product in the diet of juvenile Megalobrama amblycephala

The paper is good and well written, also it contains useful information for the readers of aquaculture sciences, but it needs some improvements

In abstract; add one sentence about your design

Line 17: should be MO2, and MO4   ….FMO2, and FMO4

Why the author analyzed the free amino acid composition of the muscle in control, MO4 AND FMO4 only? Where MO2 and FMO2?

TABLE 3: Where is the p value of IBW and FBW?

The paper is good and well written

Reviewer 2 Report

The study conducted by Jiang et al. is good and valuable for readers and researchers. However, many changes should be made before further processing.

The manuscript needs to be revised for language.

The abstract lacks the methods and analysis measured.

Introduction:

L42-50: not suitable for this study. There was no replacement with fishmeal in this study; replacement is difficult due to the significant difference in protein content between fishmeal and moringa.

L71 and others: at the first mention, add the common name and Latin name of fish spp. Please revise all fish spp.

L81: missing reference

The introduction didn’t move uniformly. There is a transition every while from subject to different.

L97 and others. The abbreviations should be defined as first mentioned. Please revise the manuscript.

The aim of the study is not precise.

Methods:

Information about the condition of fermentation (temperature, PH, how to control fermentation, the amount of moringa and bacteria,.etc.) and how it was done should be added.

Experimental diets should be explained in detail.

L137: experimental diet?

How was the fish fed?

Measurement of growth performance parameters was missed. How were they calculated? The growth parameters should be separated from sampling in a separate subsection.

L150-153: how many samples were taken?

L153: why did the authors measure the free amino acids in such three groups?

Please put each analysis in a separate subsection for clarification.

L177: what were the samples collected (which organs) for each gene? How many samples? Amount from each sample? Method of collection and preservation of these samples?

Results:

Statistical analysis is inappropriate; the data should be analyzed only with ONE-WAY ANOVA, and there is no need for TWO-WAY ANOVA. Please reanalyze and rewrite the text accordingly. The authors should add the P-value for each parameter measured. Please use Tukey’s test for comparing the means, as it is more realistic than Duncan’s test.

L208 is the first time to mention the common name of the fish used in the experiment. Please add it before the Latin name at first mentioned.

L260, 261:” enhanced and improvement” are unsuitable for GLU and TG as their increase is not preferable.

Please define any abbreviation in tables and figures in the footnote of the tables and legends of figures.

Discussion:

L350-352: what is its relation to the sequence? Please delete this sentence.

L367-370: rewrite this sentence.

The discussion needs more proper explanations of the results.

The conclusion is very short. It should be supplied with numerical data, the mechanism of occurring these results, recommendations, optimal level and form, future ideas, and limitations.

The study conducted by Jiang et al. is good and valuable for readers and researchers. However, many changes should be made before further processing.

The manuscript needs to be revised for language.

The abstract lacks the methods and analysis measured.

Introduction:

L42-50: not suitable for this study. There was no replacement with fishmeal in this study; replacement is difficult due to the significant difference in protein content between fishmeal and moringa.

L71 and others: at the first mention, add the common name and Latin name of fish spp. Please revise all fish spp.

L81: missing reference

The introduction didn’t move uniformly. There is a transition every while from subject to different.

L97 and others. The abbreviations should be defined as first mentioned. Please revise the manuscript.

The aim of the study is not precise.

Methods:

Information about the condition of fermentation (temperature, PH, how to control fermentation, the amount of moringa and bacteria,.etc.) and how it was done should be added.

Experimental diets should be explained in detail.

L137: experimental diet?

How was the fish fed?

Measurement of growth performance parameters was missed. How were they calculated? The growth parameters should be separated from sampling in a separate subsection.

L150-153: how many samples were taken?

L153: why did the authors measure the free amino acids in such three groups?

Please put each analysis in a separate subsection for clarification.

L177: what were the samples collected (which organs) for each gene? How many samples? Amount from each sample? Method of collection and preservation of these samples?

Results:

Statistical analysis is inappropriate; the data should be analyzed only with ONE-WAY ANOVA, and there is no need for TWO-WAY ANOVA. Please reanalyze and rewrite the text accordingly. The authors should add the P-value for each parameter measured. Please use Tukey’s test for comparing the means, as it is more realistic than Duncan’s test.

L208 is the first time to mention the common name of the fish used in the experiment. Please add it before the Latin name at first mentioned.

L260, 261:” enhanced and improvement” are unsuitable for GLU and TG as their increase is not preferable.

Please define any abbreviation in tables and figures in the footnote of the tables and legends of figures.

Discussion:

L350-352: what is its relation to the sequence? Please delete this sentence.

L367-370: rewrite this sentence.

The discussion needs more proper explanations of the results.

The conclusion is very short. It should be supplied with numerical data, the mechanism of occurring these results, recommendations, optimal level and form, future ideas, and limitations.

Author Response

Dear Editor,

We appreciate the professional comments from the reviewers (fermentation-2406974). Please check our point-to-point responses and the revisions (yellow highlighted in the revised manuscript). I hope we have adequately addressed the comments and suggestions from reviewers. Please feel free to contact me if any further improvements to the manuscript are deemed necessary.

Best wishes

Linghong Miao,

Wuxi Fisheries College, Nanjing Agricultural University, Wuxi 214081, China

The first author: Wenqiang Jiang

Wuxi Fisheries College, Nanjing Agricultural University, Wuxi 214081, China

Reviewers' comments:

Reviewer #2: The study conducted by Jiang et al. is good and valuable for readers and researchers. However, many changes should be made before further processing.

The abstract lacks the methods and analysis measured.

Response: Thanks for your correction. We have revised accordingly. (highlighted in YELLOW in the revised manuscript, lines 17-22).

Abstract: This study assessed the potential applicability of Moringa oleifera leaf meal (MO) and fermented Moringa oleifera leaf meal (FMO) as feed supplements for aquatic animals. Five experimental diets, including the basal diet (control), 2.2% and 4.4% MO supplemented diets (MO2, and MO4), and 2.2% and 4.4% FMO supplemented diets (FMO2, and FMO4), were prepared for feeding juvenile blunt snout bream (Megalobrama amblycephala). After the eight weeks feeding trial, the growth performance, muscle nutritional composition, plasma and hepatic biochemistry indicators were measured. The results demonstrated that MO and FMO had no detrimental effects on the growth performance of M. amblycephala. The muscle crude protein, crude lipid, and total free amino acids content were significantly enhanced in the FMO4 group (P<0.05)……

L42-50: not suitable for this study. There was no replacement with fishmeal in this study; replacement is difficult due to the significant difference in protein content between fishmeal and moringa.

Response: Thanks for your suggestion. We have revised accordingly. (highlighted in YELLOW in the revised manuscript, lines 38-46).

In recent years, the resource shortage leads to elevated prices of fishmeal, soybean meal, cottonseed meal in aquatic feed, as well as the other protein sources [1]. To alleviate the phenomenon of competition between humans and animals for food, some non-food protein sources obtained widely concerned, such as mycoprotein (Clostridium autoethanogenum protein, yeast hydrolysate) [2,3], insect protein (Tenebrio molitor meal, Hermetia illucens meal) [4,5] and plant leaf protein (mulberry leaf meal, Hygrophila spinosa leaf meal) [6,7]. In another aspect, some processing technology, such as enzymatic digestion [8] and fermentation [6,9], were also used for improving the quality and utilization efficiency of raw feed ingredients by aquatic animals.

References

  1. FAO fisheries and aquaculture information and statistics services. In: Aquaculture Production: Quantities, pp. 1950–2016.; 2020.
  2. Dai, J.; Chen, T.; Guo, X.; Dai, Z.; He, Z.; Hu, Y. Evaluation of fish meal replacement by Clostridium autoethanogenum protein in diets for juvenile red swamp crayfish (Procambarus clarkii). Aquaculture. 2023, 570, 739379.
  3. Yuan, X. Y.; Liu, W. B.; Liang, C.; Sun, C. X.; Xue, Y. F.; Wan, Z. D.; Jiang, G. Z. Effects of partial replacement of fish meal by yeast hydrolysate on complement system and stress resistance in juvenile Jian carp (Cyprinus carpio Jian). Fish Shellfish Immunol. 2017, 67, 312-321.
  4. Li, H.; Hu, Z.; Liu, S.; Sun, J.; Ji, H. Influence of dietary soybean meal replacement with yellow mealworm (Tenebrio molitor) on growth performance, antioxidant capacity, skin color, and flesh quality of mirror carp (Cyprinus carpio specularis). Aquaculture. 2022, 561, 738686.
  5. Huang, B.; Zhang, S.; Dong, X.; Chi, S.; Yang, Q.; Liu, H.; Xie, S. Effects of fishmeal replacement by black soldier fly on growth performance, digestive enzyme activity, intestine morphology, intestinal flora and immune response of pearl gentian grouper (Epinephelus fuscoguttatusâ™€× Epinephelus lanceolatus♂). Fish Shellfish Immunol. 2022, 120, 497-506.
  6. Jiang, W.; Lin, Y.; Qian, L.; Miao, L.; Liu, B.; Ge, X.; Shen, H. Mulberry leaf meal: A potential feed supplement for juvenile Megalobrama amblycephala “Huahai No. 1”. Fish Shellfish Immunol. 2022, 128, 279-287.
  7. Maiti, M. K.; Sahu, N. P.; Sardar, P.; Shamna, N.; Deo, A. D.; Gopan, A.; Sahoo, S. Optimum utilization of Hygrophila spinosa leaf meal in the diet of Labeo rohita (Hamilton, 1822) fingerlings. Aquac Rep. 2019, 15, 100213.
  8. Poolsawat, L.; Yang, H.; Sun, Y. F.; Li, X. Q.; Liang, G. Y.; Leng, X. J. Effect of replacing fish meal with enzymatic feather meal on growth and feed utilization of tilapia (Oreochromis niloticus× O. aureus). Anim Feed Sci Technol. 2021, 274, 114895.
  9. Zhang, X.H.; Sun, Z.Y.; Cai, J.F.; Wang, J.H.; Wang, G.B.; Zhu, Z.L.; et al. Effects of dietary fish meal replacement by fermented moringa (Moringa oleifera) leaves on growth performance, nonspecific immunity and disease resistance against Aeromonas hydrophila in juvenile gibel carp (Carassius auratus gibelio var. CAS III). Fish Shellfish Immunol. 2020, 102, 430-439.

L71 and others: at the first mention, add the common name and Latin name of fish spp. Please revise all fish spp.

Response: Thanks for your kind reminder. We have revised the whole manuscript accordingly.  Nile tilapia (Oreochromis niloticus), sea bass (Lateolabrax japonicus), African catfish (Clarias gariepinus), gibel carp (Carassius auratus gibelio) et al.

L81: missing reference

Response: Thanks for your correction. We have revised accordingly. (highlighted in YELLOW in the revised manuscript, lines 77-78).

Furthermore, fermentation products promote the growth performance of aquatic animals by affecting the intestinal environment [26, 27].

References

  1. Kari, Z.A.; Kabir, M.A.; Dawood, M.A.; Razab, M.K.A.A.; Ariff, N.S.N.A.; Sarkar, T.; et al. Effect of fish meal substitution with fermented soy pulp on growth performance, digestive enzyme, amino acid profile, and immune-related gene expression of African catfish (Clarias gariepinus). Aquaculture. 2022, 546, 737418.
  2. He, Y.F.; Guo, X.W.; Tan, B.P.; Dong, X.H.; Yang, Q.H.; Liu, H.Y.; et al. Replacing fish meal with fermented rice protein in diets for hybrid groupers (Epinephelus fuscoguttatusâ™€× Epinephelus lanceolatus♂): Effects on growth, digestive and absorption capacities, inflammatory-related gene expression, and intestinal microbiota. Aquac Rep. 2021, 19, 100603.

The introduction didn’t move uniformly. There is a transition every while from subject to different.

Response: Thanks for your suggestion. We have revised accordingly. (highlighted in YELLOW in the revised manuscript, lines 83-84).

MO has a hypolipidemic effect by inhibiting cholesterol synthesis and reducing hepatic lipid vacuolation in mono-sex tilapia (Oreochromis niloticus).

L97 and others. The abbreviations should be defined as first mentioned. Please revise the manuscript.

Response: Thanks for your correction. We have revised the whole manuscript accordingly. (highlighted in YELLOW in the revised manuscript, lines 93).

1,1-diphenyl-2-picrylhydrazyl (DPPH).

The aim of the study is not precise.

Response: Thanks for your correction. We have revised accordingly. (highlighted in YELLOW in the revised manuscript, lines 96-99).

Therefore, this research was conducted to study the effects of MO and its fermentation product of FMO on growth performance, feed utilization, and hepatic antioxidant capacity as aquatic animal feed supplements of Megalobrama amblycephala.

Information about the condition of fermentation (temperature, PH, how to control fermentation, the amount of moringa and bacteria,.etc.) and how it was done should be added.

Response: Thanks for your kind reminder. We have revised accordingly. (highlighted in YELLOW in the revised manuscript, lines 109-114).

  1. oleifera leaves were purchased from Greenway Agriculture Co., Ltd. (Zhongshan, China). MO was obtained by crushing M. oleifera leaves and removing solid impurities at 60 mesh. FMO was prepared using Bacillus subtilis SIX-15 (Table S1 & Figure S1). Firstly, Bacillus subtilis SIX-15 was incubated at 35°C for 18-20h in Luria-Bertani broth medium. The fermentation substrate was prepared according to the mass ratio of MO: sterilized water = 1:9. Bacillus subtilis SIX-15 was inoculated into sterilized M. oleifera leaf mixture to 2×105 cfu/mL and fermented for 48 h at 35°C. Thereafter, FMO was prepared after being freeze-dried.

Experimental diets should be explained in detail.

Response: Thanks for your kind reminder. We have revised accordingly. (highlighted in YELLOW in the revised manuscript, lines 118-120).

All ingredients were sieved to remove any solid impurities and then thoroughly mixed. Afterward, soybean oil and water were gradually added to produce a sinking pellet (2 mm) using an F-26(II) pelletizer (South China University of Technology, China).

L137: experimental diet?

Response: Thanks for your question. We have revised accordingly. (highlighted in YELLOW in the revised manuscript, lines 143-144).

The control, MO2, MO4, FMO2, and FMO4 diets were randomly assigned to three floating cages.

How was the fish fed?

Response: Thanks for your question.

During the feeding trial, the fish were hand-fed carefully three times daily at 7:30, 11:30, and 17:30 until apparent satiation (based on visual observation) for 56 days. (highlighted in YELLOW in the revised manuscript, lines 144-146).

Measurement of growth performance parameters was missed. How were they calculated? The growth parameters should be separated from sampling in a separate subsection.

Response: Thanks for your question. We have revised accordingly. (highlighted in YELLOW in the revised manuscript, lines 165-172).

L150-153: how many samples were taken?

Response: Thanks for your question. We have revised accordingly. (highlighted in YELLOW in the revised manuscript, lines 154-163).

Four fish were then randomly collected from each cage for sampling. Blood of four fish were obtained using disposable medical syringes from the caudal vein. The supernatant plasma was collected after centrifuging at 4 °C and 4000 r/min for 10 min. The plasma was kept at -20°C until determined biochemical parameters. Three fish were randomly selected from each cage and immediately dissected to collect liver tissue and dorsal muscle. Liver samples were divided into two parts, that one part stored at -20°C for antioxidant parameters assay and the other stored at -80°C for genes relative expressions assay. Dorsal muscle samples from all the groups were stored at -20°C for muscle composition determination. Dorsal muscle from the control, MO4, and FMO4 groups were collected, and stored at 4°C for determining the free amino acid content.

L153: why did the authors measure the free amino acids in such three groups?

Response: Thanks for your question. In previous studies, a consistent trend in muscle crude protein content and total free amino acid content was found in freshwater fishes (gibel carp (Carassius auratus gibelio), grass carp (Ctenopharyngodon idellus), tilapia (Oreochromis niloticus) et al.) [1-3]. In the present study, the protein efficiency ratio was higher in the MO4 and FMO4 groups than in the other three groups. Similarly the muscle crude protein content showed significantly higher in the control, MO4 and FMO4 groups than in the MO2 and FMO2 groups. This study was conducted to verify the feasibility of using Moringa oleifera leaf meal (MO) and fermented Moringa oleifera leaf meal (FMO) as feed supplements for juvenile Megalobrama amblycephala. Thus, we determined the muscle free amino acids profile of fish fed diets supplementing higher inclusion level of 4.4% MO/ FMO, and compared them with that of control group.

References

  1. Xu, S. D.; Zheng, X.; Dong, X. J.; Ai, Q. H.; & Mai, K. S. Beneficial effects of phytase and/or protease on growth performance, digestive ability, immune response and muscle amino acid profile in low phosphorus and/or low fish meal gibel carp (Carassius auratus gibelio) diets. Aquaculture. 2022. 555, 738157.
  2. Tie, H. M.; Wu, P.; Jiang, W. D.; Liu, Y.; Kuang, S. Y.; Zeng, Y. Y.; Feng, L. Dietary nucleotides supplementation affect the physicochemical properties, amino acid and fatty acid constituents, apoptosis and antioxidant mechanisms in grass carp (Ctenopharyngodon idellus) muscle. Aquaculture. 2019. 502, 312-325.
  3. He, J.; Feng, P.; Lv, C.; Lv, M.; Ruan, Z.; Yang, H.; Wang, R. Effect of a fish–rice co-culture system on the growth performance and muscle quality of tilapia (Oreochromis niloticus). Aquac Reps. 2020, 17, 100367.

Please put each analysis in a separate subsection for clarification.

Response: Thanks for your kind reminder. We have revised accordingly. (highlighted in YELLOW in the revised manuscript, lines 164, 165, 173, 186, 191, 198).

2.5 Laboratory analysis

2.5.1 Growth performance

2.5.2 Feed and muscle nutrient composition.

2.5.3 Plasma biochemical parameters.

2.5.4 Hepatic antioxidant capacity

2.5.5 Real-time PCR (qPCR) analysis on genes relative expressions

L177: what were the samples collected (which organs) for each gene? How many samples? Amount from each sample? Method of collection and preservation of these samples?

Response: Thanks for your question. We have revised accordingly. (highlighted in YELLOW in the revised manuscript, lines 154-163).

Four fish were then randomly collected from each cage for sampling. Blood of four fish were obtained using disposable medical syringes from the caudal vein. The supernatant plasma was collected after centrifuging at 4 °C and 4000 r/min for 10 min. The plasma was kept at -20°C until determined biochemical parameters. Three fish were randomly selected from each cage and immediately dissected to collect liver tissue and dorsal muscle. Liver samples were divided into two parts, that one part stored at -20°C for antioxidant parameters assay and the other stored at -80°C for genes relative expressions assay. Dorsal muscle samples from all the groups were stored at -20°C for muscle composition determination. Dorsal muscle from the control, MO4, and FMO4 groups were collected, and stored at 4°C for determining the free amino acid content.

Statistical analysis is inappropriate; the data should be analyzed only with ONE-WAY ANOVA, and there is no need for TWO-WAY ANOVA. Please reanalyze and rewrite the text accordingly. The authors should add the P-value for each parameter measured. Please use Tukey’s test for comparing the means, as it is more realistic than Duncan’s test.

Response: Thanks for your kind correction. We have revised accordingly.(lines 217-328).

L208 is the first time to mention the common name of the fish used in the experiment. Please add it before the Latin name at first mentioned.

Response: Thanks for your kind reminder. We have revised the whole manuscript accordingly. (highlighted in YELLOW in the revised manuscript, lines 228, 244, 260).

Table 3. Growth performance of juvenile blunt snout bream (Megalobrama amblycephala) fed with the experimental diets

L260, 261:” enhanced and improvement” are unsuitable for GLU and TG as their increase is not preferable.

Response: Thanks for your correction. We have revised accordingly. (highlighted in YELLOW in the revised manuscript, lines 272-275).

The TG content was significantly increased in the FMO2 group versus the control group (P<0.05). Moreover, a distinct increasement in TG content was observed in the FMO2 and FMO4 groups relative to the MO4 group (P<0.05).

Please define any abbreviation in tables and figures in the footnote of the tables and legends of figures.

Response: Thanks for your kind reminder. We have revised accordingly. (lines 214-216, 232-233, 278-281, 292-295, 306-310, 325-328).

L350-352: what is its relation to the sequence? Please delete this sentence.

Response: Thanks for your question. We have revised accordingly.

L367-370: rewrite this sentence.

Response: Thanks for your correction. We have revised accordingly. (highlighted in YELLOW in the revised manuscript, lines 366-368).

Studies have found that increased levels of LDL-C, TG, and GLU in the blood promoted lipid deposition in the muscle of golden pompano (Trachinotus ovatus) and pacific abalone (Haliotis discus hannai).

The discussion needs more proper explanations of the results.

Response: Thanks for your correction. We reorganized the discussion section after performing Tukey multiple comparisons on the experimental data. (highlighted in YELLOW in the revised manuscript, lines 329-436).

The conclusion is very short. It should be supplied with numerical data, the mechanism of occurring these results, recommendations, optimal level and form, future ideas, and limitations.

Response: Thanks for your kind reminder. We have revised accordingly.

The study revealed that dietary supplementation with 2.2% and 4.4% of M. oleifera leaf meal or fermented M. oleifera leaf meal had no negative effect on the growth performance of M. amblycephala, dietary 2.2% and 4.4% fermented M. oleifera leaf meal enhanced anti-inflammatory capacity of the liver. Dietary 4.4% fermented M. oleifera leaf meal supplementation was recommend owing to the fact that it increased the muscle nutrient composition, hepatic antioxidant and anti-inflammatory capacities of the M. amblycephala. Based on the results of our present study, probiotics fermentation improved the physiological status of M. amblycephala fed M. oleifera leaf meal. It is necessary to conduct further investigations into the mechanisms by which fermented protein sources enhance nutrient deposition in aquatic animals.

Round 2

Reviewer 2 Report

Thank you for the revision. The authors improved the manuscript as suggested by the reviewer.

The authors should add the experimental design and clarify the experimental diets in the methods section " experimental diets subsection" as that mentioned in the abstract

Fine

Author Response

Reviewer #1: The authors should add the experimental design and clarify the experimental diets in the methods section " experimental diets subsection" as that mentioned in the abstract.

Response: Thanks for your correction. We have revised accordingly. (highlighted in BLUE in the revised manuscript, lines 116-121).

Five iso-nitrogenous and iso-energetic diets were designed in this experiment, including the basal diet (control), MO2 diet (2.2% MO was added to the basal diet in place of 1.1% cottonseed meal), MO4 diet (4.4% MO was added to the basal diet in place of 2.2% cottonseed meal), FMO2 diet (2.2% FMO was added to the basal diet in place of 1.1% cottonseed meal), and FMO4 diet (4.4% FMO was added to the basal diet in place of 2.2% cottonseed meal).